# Speech Enhancement Using Generative Adversarial Network by Distilling Knowledge from Statistical Method

**Jianfeng Wu, Yongzhu Hua, Shengying Yang, Hongshuai Qin and Huibin Qin ***

The Institute of Electron Device & Application, Hangzhou Dianzi University, Hangzhou 310018, China
* Correspondence: qhb@hdu.edu.cn; Tel.: +86-139-5807-3660; Fax: +86-571-86878618

**Abstract:** This paper presents a new deep neural network (DNN)-based speech enhancement algorithm by integrating the distilled knowledge from the traditional statistical-based method. Unlike the other DNN-based methods, which usually train many different models on the same data and then average their predictions, or use a large number of noise types to enlarge the simulated noisy speech, the proposed method does not train a whole ensemble of models and does not require a mass of simulated noisy speech. It first trains a discriminator network and a generator network simultaneously using the adversarial learning method. Then, the discriminator network and generator network are re-trained by distilling knowledge from the statistical method, which is inspired by the knowledge distillation in a neural network. Finally, the generator network is fine-tuned using real noisy speech. Experiments on CHiME4 data sets demonstrate that the proposed method achieves a more robust performance than the compared DNN-based method in terms of perceptual speech quality.

**Keywords:** speech enhancement; deep neural network; generative adversarial network; distill knowledge

## 1. Introduction

Single-channel speech enhancement has been studied for decades, while it is still a challenging problem in numerous application systems such as automatic speech recognition (ASR), hearing aids and hands-free mobile communication. Traditional speech enhancement algorithms are usually based on a statistical method, and consist of noise spectrum estimation and speech signal estimation [1–3]. The performance of these methods relies heavily on accurate noise estimation. However, due to the statistical assumption of the speech and noise, most of the traditional methods often fail to track non-stationary noise and lead to a high level of musical noise artifacts.

In recent years, numerous data-driven methods have been proposed to circumvent the assumption of specific distortion for the speech and noise processes [4,5]. One of the notable algorithms is the regression approach to speech enhancement based on deep neural networks (DNN), which is inspired by the successful introduction of DNN to acoustic modeling in ASR system [6]. Another kind of popular method is using the DNN to estimate an ideal binary mask (IBM) or a smoothed ideal ratio mask (IRM) in the frequency domain, which is derived from computational auditory scene analysis for monaural speech separation [7,8]. Nevertheless, all these methods only enhance the speech magnitude spectrum, leaving the phase spectrum unprocessed.

More recently, an end-to-end speech enhancement method is proposed using generative adversarial networks (GAN) [9], namely, speech enhancement GAN (SEGAN). The SEGAN, which is an end-to-end system, operates speech signal on the raw waveform, other than on the spectral domain or on some higher-level domain. The experimental results show that the GAN-based method achieves

better performance than traditional methods both in objective and subjective evaluations. Yet, all the DNN-based approaches are data-hungry: The more data is available, the better performance they gain. In the absence of more data, researchers usually train many different models on the same data and then average their predictions. Whereas, using a whole ensemble of models is too computationally expensive. To overcome this limitation, a compression method is proposed to distill the knowledge from an ensemble of neural network models [10]. This encourages us to distill knowledge from the traditional minimum mean squared error (MMSE) method, e.g., the optimally-modified log-spectral amplitude (OMLSA) speech estimator and minima controlled recursive averaging (MCRA) noise estimation approach for robust speech enhancement [2], denoted as OMLSA method.

In this paper, we propose a novel DNN-based speech enhancement method using adversarial training. The main challenge of applying DNN to speech enhancement is how to get more training pairs, especially for real data cases, where the ground-truth clean speech is not available. We use the traditional statistical-based speech enhancement method to estimate the speech signal in the real case, so that we can obtain more training pairs for the DNN. That is, we train a DNN-based model by distilling knowledge from the traditional statistical-based method. Moreover, in the view of machine learning methodology, while most of the existing DNN-based methods are trained using supervised approach [6–9], the proposed method is trained in a semi-supervised manner.

The rest of the paper is organized as follows. In Section 2, we briefly introduce the related work. Section 3 presents the proposed new speech enhancement method. A set of experimental results are provided in Section 4. Finally, we summarize our work in Section 5.

## 2. Related Work

### 2.1. Speech Enhancement Using Generative Adversarial Networks

GANs [11,12] are generative models that train two DNN models simultaneously: a generator G that captures the training data distribution, and a discriminator D that estimates the probability that a sample came from the training data rather than the generator. The SEGAN [12] uses the training pairs to train an end-to-end generator *G*. And the corresponding discriminator *D*. In the structure of SEGAN, the input of the *G* is the noisy speech, and the expected output of the *G* is the clean speech. Thus, the generator *G* performs the speech enhancement.

In the training stage of SEGAN, the inputs of *G* are the noisy speech signal $\widetilde{\mathbf{x}}$, and the latent representation $\mathbf{z}$. The output of G is the enhanced speech $\hat{\mathbf{x}} = G(\widetilde{\mathbf{x}})$. Theoretically, the training of G can be formulated as a minimization of the following loss function [9]:

$$\min_G V(G) = \frac{1}{2}\mathbb{E}_{\mathbf{z}\sim p_{\mathbf{z}}(\mathbf{z}),\widetilde{\mathbf{x}}\sim p_{\text{data}}(\widetilde{\mathbf{x}})}\left[(D(G(\mathbf{z},\widetilde{\mathbf{x}}),\widetilde{\mathbf{x}}) - 1)^2\right] + \lambda\|G(\mathbf{z},\widetilde{\mathbf{x}}) - \mathbf{x}\|_1 \tag{1}$$

where the additional $L_1$ term is minimizing the distance between the generated speech and the real clean speech $\mathbf{x}$, and $\lambda$ is the weight.

Figure 1 demonstrates the complete adversarial training process, which is a two-player game between *G* and *D*: First, *D* backpropagates the real clean speech. Then, *D* backpropagates the generated speech that comes from *G* and classifies it. Finally, the parameters of *D* are frozen and *G* backpropagates to make *D* misclassify.

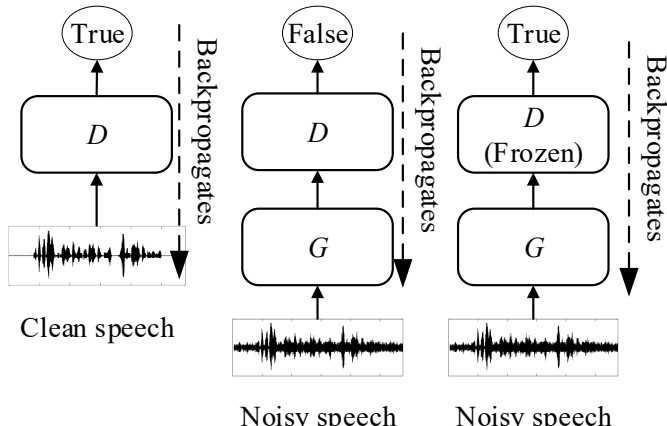

**Figure 1.** Adversarial training for speech enhancement.

First, $D$ back-props the clean speech and the generated speech to classifies them. Then, the parameters of $D$ are frozen and $G$ back-props to make $D$ misclassify.

### 2.2. Statistical-Based Speech Enhancement

The traditional statistical-based speech enhancement method operates in short time Fourier transform (STFT) domain. Let's denote $\sigma_X^2(t,k)$, $\sigma_S^2(t,k)$ and $\sigma_N^2(t,k)$ for power spectral density (PSD) of noisy speech, clean speech and noise signal, respectively. The relationship of these PSD is:

$$\sigma_X^2(t,k) = \sigma_S^2(t,k) + \sigma_N^2(t,k) \tag{2}$$

where $t$ and $k$ are the time frame index and the frequency bin index, respectively.

Figure 2 illustrates the process of typical statistical-based speech enhancement. Firstly, the noisy speech is transformed into the time-frequency domain. Then, an estimation of $\sigma_N^2(t,k)$ is obtained by means of a noise estimator, such as the improved minima controlled recursive averaging (IMCRA) [2]. After that, the speech spectrum is estimated by speech estimator, e.g., the minimum mean square error (MMSE). Finally, inverse short time Fourier transform (ISTFT) is applied to get the estimated time-domain speech signal. It should be noticed that the phase spectrum of the noisy speech is not being processed.

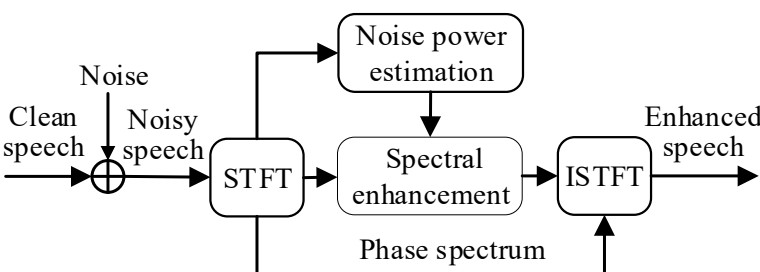

**Figure 2.** The statistical-based speech enhancement algorithm.

## 3. Adversarial Network with Distilled Knowledge from Statistical Method

### 3.1. Proposed Architecture

In this paper, we use the SEGAN framework discussed above to train an end-to-end generative DNN model for speech enhancement. The DNN model not only learns a map function from the simulated noisy speech to the ground-truth clean speech but also distill knowledge from the MMSE-based speech enhancement algorithm, which is able to de-noise real noisy speech. Figure 3

shows the framework of the proposed new method. It consists of three steps: initialization step, distilling step, and fine-tuning step. The details of the training procedures are presented in the following sections.

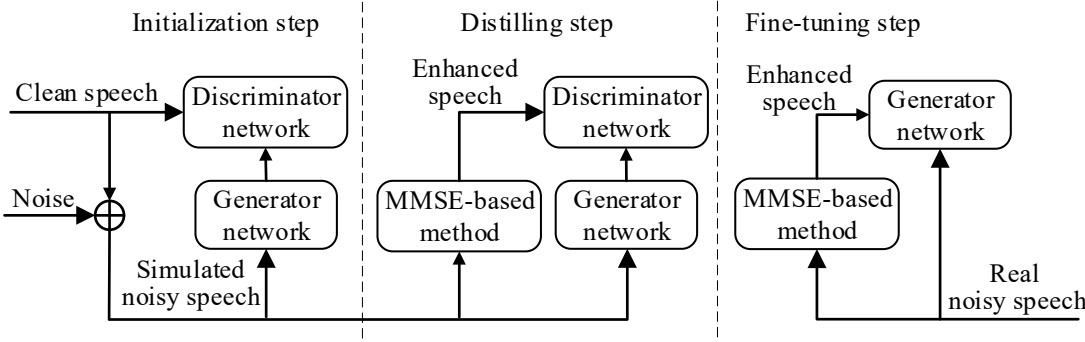

**Figure 3.** The framework of the proposed algorithm.

### 3.2. Network Initialization Using Adversarial Learning

In the beginning, the simulated noisy speech is obtained by adding various noises to clean speech. That is, the ground-truth clean speech is known. Using the simulated noisy speech and clean speech as training data pair, we train the discriminator network and the generator network simultaneously using the adversarial learning method. The network structure of the discriminator and generator is presented in Section 4.

### 3.3. Distilling Knowledge from MMSE-Based Speech Enhancement Algorithm

For the DNN-based speech enhancement method, the more the simulated speech available, the better the performance that can be obtained. In order to obtain a better generalization performance, the authors in [9] use 104 types of noise and 6 levels of signal-to-noise ratios (SNRs) simulated noisy speech to train the DNN. Yet, those simulated noisy speech cannot cover all the real cases.

To improve the generalization performance of the generator network trained in the previous step, we distill knowledge from the traditional MMSE-based speech enhancement algorithm, other than building a more complex training set. We re-use the simulated noisy speech in the previous step, and feed the data to the MMSE-based speech enhancement method, e.g., OMLSA [13] which is one of the state-of-art MMSE-based method, to get the estimated clean speech, i.e., the enhanced speech. Then, the estimated clean speech and noisy speech constitute the new training data pair. In the view of machine learning, the procedure of obtain the new training pairs can be thought of as data augmentation. Subsequently, we re-train the discriminator network and the generator network with the weights of the previous step.

This step is inspired by the knowledge distillation in a neural network [10], which uses a big model (i.e., the teacher) to teach a small model (i.e., the student) in the absence of more training data. Given the training data, the teacher model can generate more "soft target" to teach the student model. And the student model can learn what the teacher model has "taught". In this paper, the MMSE-based speech enhancement algorithm is the teacher, the DNN is the student. Given the simulated noisy speech, the DNN can learn the intrinsic mapping function of the MMSE-based method.

### 3.4. Fine-Tuning the Generator Network for Speech Enhancement

In the final step, we use real noisy speech to fine-tune the generator network trained in previous step. The discriminator network dose not participate in this training step, as we only use the generator network to perform speech estimation in the enhancement stage. For the real noisy speech, the ground-truth clean speech is not available. We use the enhanced speech, which is estimated by the OMLSA-MCRA, and the noisy speech to constitute the new training data pair.

In the view of machine learning, the fine-tune step is a transfer learning procedure. Based on those previous steps, the generator network has learned high level features of the raw speech signal using the simulated noisy speech. Yet, an obvious drawback of all previous step is that the simulated noisy speech cannot cover all the real cases, such as noise types, signal-to-noise ratios and additive noise assumption. The fine-tuning step is an adaptation process of some specific environments for the real application.

## 4. Experimental Results

### 4.1. Setup

We perform speech enhancement using the CHiME4 [14] data corpus, other than using the TIMII data corpus and noise to constitute the noisy speech, because there is no real noisy data for the constituted simulated noisy speech in TIMII data corpus. The corpus consists of real and simulated audio data taken from the 5k WSJ0-Corpus with four different types of noise, i.e., bus (BUS), cafe (CAF), pedestrian area (PED), and street junction (STR). There are totally 8738 utterances for training, 3280 utterances for validation, and 2640 utterances for testing. Besides, the corpus is recorded by a six-channel microphone array, although we only use the single-channel subset, because this paper only studies single-channel speech enhancement algorithm.

The setting for the network and the training parameters are as follows. The generator G uses an auto-encoder architecture with skip connections from the encoder to the decoder. The encoder is composed of 22 one-dimensional convolutional layers of filter width 31 and strides 2, the decoder is a mirroring of the encoder with the same parameters set. The discriminator D follows the same one-dimensional convolutional structure as the encoder of G. The weights of all layers are initialized by a Xavier initializer, and all the biases are initialized with zeros. An RMSprop optimizer with a fixed learning rate sets to 0.0002 is adopted to train the models. In order to minimize the distance between the generated speech and the real clean speech, $L_1$ term is used as a regularization [9], and the weight parameter $\lambda$ is set to 100. We carry out the training and evaluation procedures on a workstation with Intel Xeon E5-2630 CPU and two GTX 1080ti GPUs.

### 4.2. Evaluation

The perceptual evaluation of speech quality (PESQ) [14], signal to distortion ratios (SDR in dB), short-time objective intelligibility (STOI) [15], and the extended STOI (eSTOI) [16] are used to evaluate the quality of the enhanced speech signal. For the simulated speech data, the clean speech is available, while for the real speech data, the ground-truth clean speech is not available. Thus, we use the close-talking microphone recordings (i.e., channel index 0) as the underlying clean speech. We use the pretrained SEGAN and the OMLSA for comparison. In addition, the noisy speech is also considered for comparison (denoted by None).

The SDR, STOI, and eSTOI scores for speech quality test on the development set and evaluation set are shown in Figure 4, in which the dev and eval are short for development and evaluation set, respectively. The results demonstrate that our approach outperforms the SEGAN and OMLSA for the simulated data, and yields comparable performance as the traditional state-of-art OMLSA algorithm for the real data. For example, in the simulated data of the development set, compared to SEGAN and OMLSA, the proposed method achieves about 26.01% and 21.10% relative gain in terms of SDR. Another example is in the real data of the evaluation set, the eSTOI score of the proposed method is 0.34 while the eSTOI score of the SEGAN and OMLSA are 0.28 and 0.31, respectively.

The results in Figure 4 are the average scores of all environment condition. It is necessary to investigate the details of the compared methods in the different types of environments, i.e., BUS, CAF, PED, and STR. In this experiment, we use PESQ as the criteria to evaluate the performance. The PESQ results in Table 1 show that, for the real data, all the compared methods get the highest scores in PED, and get the lowest scores in the BUS. We have listened to and analyzed the recorded audio. This is due

to the bus environment being much more noisy than the pedestrian area. From the results, we can conclude that the proposed method is not sensitive to noise types, and is more robust than the SEGAN.

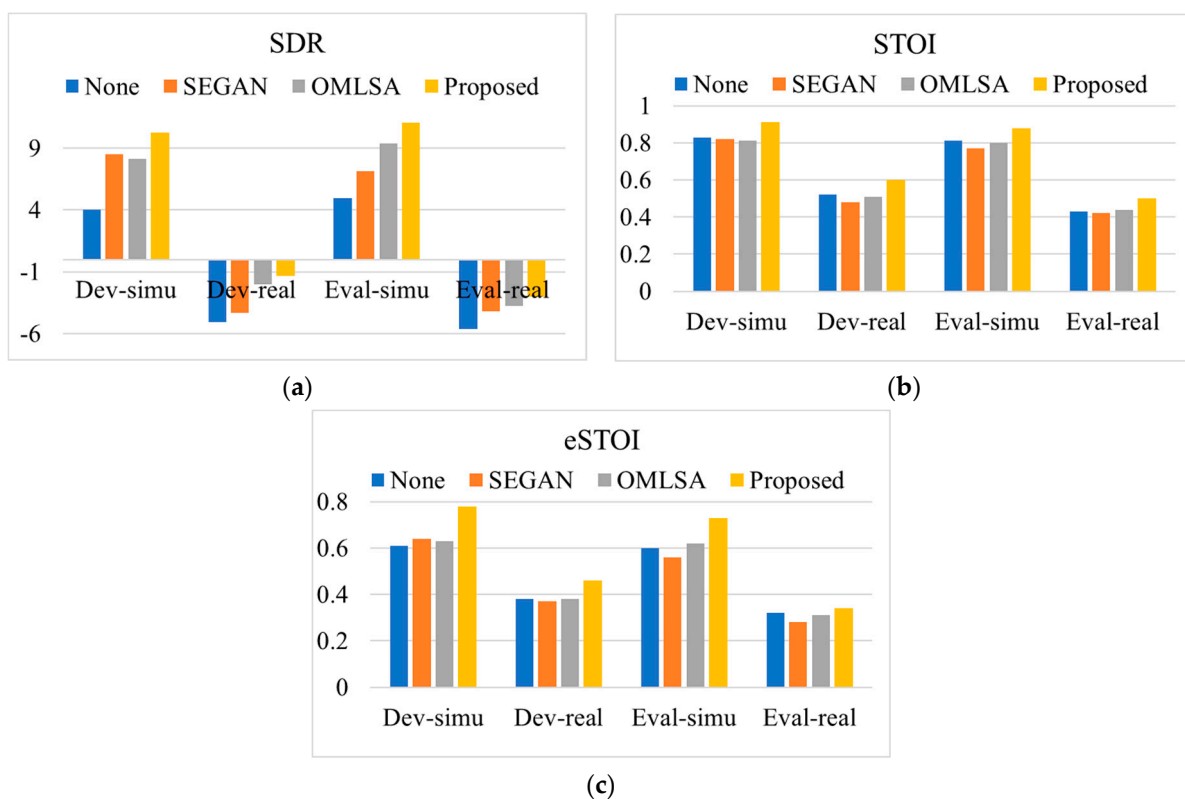

**Figure 4.** SDR (dB), eSTOI and STOI scores for speech quality test on the development set and evaluation set. (**a**) SDR (dB), (**b**) STOI, (**c**) eSTOI.

**Table 1.** PESQ scores for speech quality test on the development set and evaluation set for each environment.

| Method | Environment | Dev-Simu | Dev-Real | Eval-Simu | Eval-Real |
|---|---|---|---|---|---|
| None | BUS | 2.15 | 2.03 | 2.16 | 2.17 |
| | CAF | 1.87 | 2.18 | 1.88 | 2.37 |
| | PED | 2.12 | 2.33 | 1.93 | 2.35 |
| | STR | 1.95 | 2.11 | 1.97 | 2.38 |
| SEGAN | BUS | 2.08 | 1.96 | 2.10 | 1.98 |
| | CAF | 1.96 | 2.01 | 1.89 | 2.26 |
| | PED | 2.03 | 2.20 | 2.03 | 2.21 |
| | STR | 1.88 | 2.03 | 1.91 | 2.32 |
| OMLSA | BUS | 2.18 | 2.06 | 2.21 | 2.19 |
| | CAF | 1.91 | 2.21 | 1.92 | 2.32 |
| | PED | 2.32 | 2.40 | 2.32 | 2.42 |
| | STR | 2.02 | 2.27 | 2.02 | 2.46 |
| Proposed | BUS | 2.20 | 2.03 | 2.32 | 2.28 |
| | CAF | 2.08 | 2.30 | 2.08 | 2.51 |
| | PED | 2.36 | 2.45 | 2.48 | 2.43 |
| | STR | 2.08 | 2.26 | 2.13 | 2.50 |

Finally, we have conducted an informal subjective preference evaluation comparison between the SEGAN enhanced, the OMLSA enhanced and the proposed enhanced speech for 100 noisy speech utterances that random picked from the evaluation set. Twelve listeners (9 male and 3 female) are

recruited for the evaluation. The tester is asked to select the one he/she prefers for each enhanced speech. Thus, there are totally $100 \times 3 \times 12 = 3600$ selections, whose statistical results are shown in Table 2. The results show that the preference ratio for SEGAN, OMLSA, and the proposed is 14.0%, 14.8%, and 71.3%, respectively. From the results, we can conclude that the proposed method is superior to the compared method in term of objective metric.

**Table 2.** Subjective preference evaluation results of the compared methods on 100 noisy speech utterances evaluation set.

| Method | Eval-Simu | Eval-Real | Ave |
|---|---|---|---|
| SEGAN | 15.3% | 12.7% | 14.0% |
| OMLSA | 12.6% | 16.9% | 14.8% |
| Proposed | 72.1% | 70.4% | 71.3% |

## 5. Conclusions

In this paper, we proposed a robust end-to-end speech enhancement method using the generative adversarial network by distilling knowledge from the statistical method. The proposed method is more efficient in that it does not need to extract features from the speech signal, train many different models on the same data, or use many noise types to enlarge the simulated noisy speech. Hence, it does not suffer from phase mismatch between underlying clean speech and estimated speech, overfitting of the simulated data, or low generalization capability, which is the main problem of DNN-based speech enhancement methods. Experiments on CHiME4 data sets demonstrated that it outperforms the OMLSA and SEGAN, especially for the real data. The new speech enhancement method may be preferred in practical applications due to its robustness.

**Author Contributions:** Conceptualization, J.W.; Methodology, J.W.; Software, J.W.; Validation, J.W., Y.S. and H.Q.; Formal analysis, Y.H.; Investigation, Y.H.; Data curation, J.W.; Writing—original draft preparation, J.W.; Writing—review and editing, H.Q.; Supervision, H.Q.; Project administration, H.Q.

**Funding:** This research received no external funding.

**Conflicts of Interest:** The authors declare no conflicts of interest regarding the publication of this paper.

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
