# Peer review of "Speech Enhancement Using Generative Adversarial Network by Distilling Knowledge from Statistical Method"

_applsci, doi:10.3390/app9163396_

Round 1
Reviewer 1 Report
At first I did not understand the term “distilled”. I am not English speaking, though. It seems to refer to “extract”? It would be more understandable for all readers, but I see it is all around the paper.
Tables are difficult to read with so many metrics, so it is not easy to compare things. A Figure would be more useful, especially in Table 2. And probably, development results could be omitted.
In the conclusions, line 189, you say “especially for the real data”. I think it should be, “especially for the simulated data”.
There is no explanation as why the results on simulated speech show a large improvement over the baselines, but very little improvement on real speech.
Are the differences significant for real speech?
Also, in the conclusions you claim that you do not need a large number of noise types to enlarge the simulated noisy speech, but in your architecture you use simulated speech. Please, explain more clearly the differences.
There are lots of English mistakes, syntax errors and words missing. Here is a list of corrections, but there are more:
14: a mountain types -> a large number of noise types
28: Traditional speech enhancement algorithms usually based on statistical -> Traditional speech enhancement algorithms are usually based on a statistical
51: the traditional the minimum -> the traditional minimum
61: the existing DNN‐based method is training in a supervised means -> the existing DNN‐based methods train in a supervised way
62: the proposed new method is training in a semi‐supervised means -> the proposed new method trains in a semi‐supervised way
67: The SEGAN [12] use -> The SEGAN [12] uses
91: obtain -> obtained
110: various noise -> various noises
117: the performance can obtain -> the performance that can be obtained
141: we only use -> although we only use
148: learning rate is set to 0.0002 is adopted -> learning rate set to 0.0002 is adopted
150: set to 100 -> is set to 100
165: in the simulated data of development set, compare with -> in the simulated data of the development set, compared to
167: evaluation set -> the evaluation set
of the proposed -> of the proposed method
177: is not sensitive noise types -> is not sensitive to noise types
184: extract feature -> extract features
Author Response
Now we have revised the paper exactly according to the reviewers’ and editor’s comments. Please see the attachment.

Reviewer 2 Report
@page { margin: 0.79in } p { margin-bottom: 0.1in; line-height: 115% }This paper presents a speech enhancement technique that is a mixture between a deep learning strategy using generative adversarial networks and a classical statistical speech enhancement approach. The training procedure consists of three stages: the first one is the habitual GAN training; the second one obtains enhanced speech using the statistical method and then retrains the generator and discriminator; and the last step is a fine-tuning of the network using enhanced speech from the previous step and the corresponding noisy signal. The method is compared to SEGAN and a statistical method on the CHiME4 dataset, and the results suggest that this method is superior in terms of speech quality according to a set of objective measures.
The proposed method has less computational cost than other techniques that train an ensemble of models for speech enhancement and, according to the authors, it requires less data. However, the method is poorly explained and some design decisions are not justified. The presented experimental results suggest that the proposed method achieves superior performance compared to others presented in the literature; no subjective evaluation has been performed though.
Some detailed comments can be found below.
Abstract:
- “use a mountain types” → not very formal
- “to enlarge the simulated noisy speech … a mass of simulated noisy speech” → this sentence is a bit repetitive
- “distilling knowledge from the statistical method” → the abstract should in--troduce how knowledge is distilled from the statistical method instead of just mentioning that.
Introduction
- “Traditional speech enhancement algorithms usually based on statistical method” → are usually based on statistical methods
- “based on deep neural networks” → add (DNN)
- “in the ASR system” → in ASR systems
- “using the generative adversarial network” → using generative adversarial networks
- “The SEGAN operates speech signal on the waveform level” → this sentence sound strange, please rewrite
- “than traditional method both in the objective evaluation and subjective evaluation” → than traditional methods both in objective and subjective evaluations
- “approaches are data-hungry,” → replace comma with :
- “to distill knowledge from the traditional the minimum mean squared error” → from the traditional minimum…
- “OMLSA method” → first you wrote OML-SA and then OMLSA, please be coherent with the abbreviations.
- “so that, we can obtain more training pairs”→ so that we can obtain more training pairs
- “Moreover, in the view of machine learning methodology, all the existing DNN based method is training in a supervised means [6-9], the proposed new method is training in a semi-supervised means.” → Moreover, while most of the existing DNN methods are trained using supervised approaches, the proposed method is trained in a semi-supervised manner
Section 2
- “The SEGAN use … end-to-end generator, and” → uses. No need for a comma there
- “The generator G performs speech enhancement” → this sentence is too tough, maybe you could explain a little bit why G is able to perform speech enhancement
- “The inputs of the G” → of G/of the generator
- “as minimizing the flowing loss function.” → as a minimization of the following loss function:
- Eq. 1 is borrowed from [9], this should be mentioned
- Figure 1 does not provide much information, either improve it (and its caption) or remove it.
- “back-props” → two informal (backpropagates)
- “and classifies them as fake” → classifies it
- “D’s parameters” → the parameters of D
- The explanation of the paragraph starting in line 79 is quite poor and can be improved.
- “where t and k is” → are
- “And then” (line 90) → Then
- “obtain by a noise estimator” → obtained by means of a noise estimator
- “e.g. the improved minima” → such as the improved minima …, to cite an example
Section 3
- “the noisy speech of real case” → real noisy speech
- “will be introduced” (line 105) → are presented/described
- “will be presented in section 4” (line 114) → is presented in Section (capital S) 4
- “the better the performance can obtain” → without “can obtain”
- Yet, the 104 types...” → no need to repeat that, just replace with “those”
- “e.g. OMLSA” → you should explain why you chose this method and not others
- “the enhanced speech and noisy speech constitute the new training pair” → I guess the discriminator has to recognized the enhanced speech as true speech, but it is not clear. Also, this step is not explained or motivated in the paper.
- “initialized weight” → weights
- “network, which is trained in previous step” → network trained in previous steps
- “The discriminator network is not used in this training step” → Please improve the explanation of this
- Again, it is not clear why you are using enhanced speech as clean speech and why this can make the system work better.
Section 4
- “other than using” → it is not clear whether you are using TIMII data set or not (I guess you mean TIMIT)
- “and 2640 utterances for the test” → for testing
- “Besides, the corpus...” → Even though the corpus… Also, why are you using one channel only?
- “of 22 one dimensional” → of 22 one-dimensional
- The parameters of the GAN are borrowed from [9], you should mention that.
- “as G’s encoder” → as the encoder of G
- “with a fixed learning rate is set to” → with a fixed learning rate equal to
- “weight parameter lambda set to” → is set to
- “We use the close-talking microphone as the underlying clean speech” → Is there any other research justifying the use of this channel as clean speech? If not, you should justify your decision
- “comparable performance compared” → please correct this
- “compare with SEGAN and” → compared with
- “Another example is in the real data of evaluation set” → is in the real data of the evaluation set
- “score of the proposed” → add “method”
- “,in comparison” → while
- Table 1 is poorly designed and the results of the different methods are difficult to compare. Consider reorganizing it a bit.
- “It’s necessary” → It is necessary
- “different type of environments” → different types of environments
- “the bus environment is much noisy” → being much more noisy
- “is not sensitive noise types” → is not sensitive to noise types
- “and more robust than the SEGAN” → and is more robust
Section 5
- “The proposed method is robust in that” → those qualities do not show robustness. You can say that the training method is more efficient, but not more robust
- “the main challenge problem of the DNN-based speech enhancement method” → the main problem of DNN-based speech enhancement methods
- “The new speech enhancement method may be preferred” → The experimental validation is not enough for stating this. For example, there is no subjective evaluation, and the computational cost of the different methods is not compared.
References
- 1. minimu → minimum
- Please check all the references, something happened with the template and the first author appears different than the others many times.
Author Response
Now we have revised the paper exactly according to the reviewers’ and editor’s comments.Please see the attachment.
